# A Data-Driven Method for Ship Collision Risk Detection in Heavy Traffic Waters

Haoran Lv
*School of Navigation*
*Wuhan University of Technology*
Wuhan, China
State Key Laboratory of
Maritime Technology and Safety
Wuhan, China
lvhr@whut.edu.cn

Junmin Mou
*School of Navigation*
*Wuhan University of Technology*
Wuhan, China
State Key Laboratory of
Maritime Technology and Safety
Wuhan, China
Moujm@whut.edu.cn

Liang Zhang
*School of Navigation*
*Wuhan University of Technology*
Wuhan, China
State Key Laboratory of
Maritime Technology and Safety
Wuhan, China
hyzhangliang@whut.edu.cn

Mengxia Li
*Intelligent Transportation System*
*Research Center Wuhan*
*University of Technology*
Wuhan, China
State Key Laboratory of
Maritime Technology and Safety
Wuhan, China
limengxia@whut.edu.cn

Pengfei Chen
*School of Navigation*
*Wuhan University of Technology*
Wuhan, China
State Key Laboratory of
Maritime Technology and Safety
Wuhan, China
Chenpf@whut.edu.cn

*Abstract*—**Intended to solve the problem of poor generalization of complex collision risk model, this paper proposes a novel collision risk model based on data-driven. Firstly, the main traffic pattern is clustered by Ordering Points to Identify the Clustering Structure algorithm. Afterwards, using probability statistics and mining the altering behaviors identify the abnormal behavior. And then, encounter situation and avoidance behaviors are identified by analyzing the relative motion characteristics of the two ships. Finally, by Kullback-Leibler dispersion and kernel density estimation methods, the key parameters are extracted from encounter data with avoidance behaviors. The collision risk model based on data-driven is constructed. To verify the effectiveness of the proposed method, the method is used in a heavy traffic area, the mouth of Yangtze River, China. The results show that the collision risk model based on data-driven can be more accurate.**

*Keywords—collision risk detection, heavy traffic waters, data-driven method, ship domain model, encounter situation*

## I. INTRODUCTION

From 2014 to 2022, ship collision is the main event type of ship accidents, with 21.6% of the occurrences [1]. In order to reduce the occurrence of ship collisions, ship collision avoidance decisions need to be studied. However, before the collision avoidance decision can be investigated, the collision risk between two ships needs to be understood. So, it is necessary to do some research on the field of ship collision risk. Currently, research on ship collision risk is divided into four categories: (1) indicator method, (2) safe boundary method, (3) velocity-based method and (4) data-driven. [2].

Nowadays, some of the more common indicator methods for determining collision risk are Collision Risk Index (CRI)

method [3-5] and fusion parameter method [6, 7]. Above methods, variables are clear and easy to understand. However, in the process of encountering, the movement state of the ship and the interrelationship with other ships change all the time. Meanwhile, some key parameters of indicator method are more subjective. So, these deficiencies in the methodology may lead to errors in judging collision risk.

One of the classical of the safety boundary methods is the ship domain approach[8-12]. The collision risk model based on ship domain sets safety field around own vessel. If another ship intrudes into the area, the collision risk is considered to be existed. However, the ship domain which is the key element in this kind of method does not change with the changing traffic environment. The generalization of this method is relatively weak.

Now, there are a certain number of velocity -based approaches to study collision risk[13-16]. By exploring the principle of avoidance operation, ship collision risk model based on velocity is used to calculate the possibility of collision accidents. However, in actual complex encounter scenarios with multifactorial influences, it's a little difficult to identify the collision risk.

The collision risk models based on data driven use neural network [17-19], support vector machine [20] and other artificial intelligence algorithm to look for patterns from historical data, so as to predict the collision risk. With amount of AIS data, we are able to fit the key parameters of the collision risk model for the specified waters. The fitted parameters make the model more accurate.

Ship automatic identification system (AIS) data contains ship's dynamic and static information, which can visually display the process of sailing. Through exploring AIS data, officer's on watch (OOW) experience also can be verified. In heavy traffic waters, the basic information and specific experience are important in studies of collision risk. This information is helpful to identify ships' regular and abnormal behaviors, reflects maneuvering conditions in the real process of avoiding collision. It can also make the parameters set of the collision risk model more appropriate for the specific waters. Thus, AIS data is important for identifying the key parameter of collision risk model based on data driven.

At present, some collision risk models based on data-driven only focus on the data itself [21-25]. However, avoidance behaviors as reflecting to the collision risk are neglected in the process of research collision risk. In order to ensure the accuracy of the model based on data-driven, avoidance behaviors also need to be study. This paper proposes a novel collision risk method based on AIS data which considers avoidance behaviors. In section II, methodology which is used to constructing collision risk model is introduced in detail. In section III, the feasibility of this model is verified by the real case. In section IV, the advantage and weakness of this model are concluded.

## II. METHODOLOGY

### A. Overview of the Study

In this study, an innovative ship collision risk models based on data-driven is proposed. In this section, the methodology of this research is described in detail, and the organizational framework of the study is shown in the following figure 1. The first part is that using Ordering Points to Identify the Clustering Structure (OPTICS) cluster the historical AIS data to mine the main traffic pattern. According to probabilistic statistics and the course change, abnormal behaviors are identified from AIS data. Based on the analysis of the relative motion characteristics of the two ships, the avoided vessel can be matched accurately and the ship encounter scenarios containing the avoidance behavior is able to be extracted. The encounter scenarios extracted improved data quality in collision risk model based on data-driven. In the second part, based on historical AIS data and extracted encounter scenarios, the Kullback-Leibler (KL) dispersion and kernel density estimation methods were used to determine the most suitable ship domain parameters for the study waters. By these key parameters, the novel collision risk model is constructed.

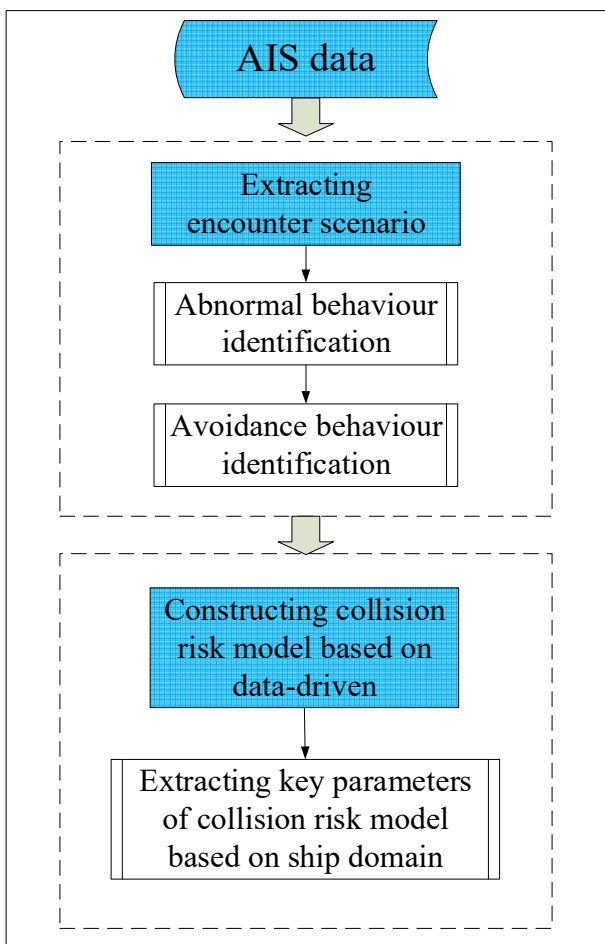

Fig.1. The overall flowchart of the research

### B. Extracting Encounter Scenario

1)abnormal behaviors identification

Firstly, in order to identify the realization of abnormal ship behavior, it is chosen to take the clustering method to identify the traffic patterns in the study waters. The scenario which is studied in this paper is the traffic intensive water, characterized by many types of ships, high traffic flow, various shapes of ship trajectories. Therefore, OPTICS, a kind of the density-based clustering algorithm, which has the advantages of noise elimination, robustness and being insensitive to parameter settings is adopted to realize the clustering of ship trajectories.

As shown in Figure 2, instead of directly generating clustering results, the OPTICS algorithm ranks the objects in the dataset, meanwhile, generates a map of the distribution of reachable distances. The formula for calculating the reachable distance is as follows:

$$\begin{cases} N_\varepsilon(p) = \{p_i \in D | dis(p_i, p_i) \le \varepsilon\} \\ |N_\varepsilon(p)| = MinPts \end{cases} \quad (1)$$

where $p_i$ means one of the samples $D$; $N_\varepsilon(p)$ means the number of $p_i$ satisfying the above equation; $\varepsilon$ is the reachability distance of sample $p_i$ when the core object is $MinPts$.

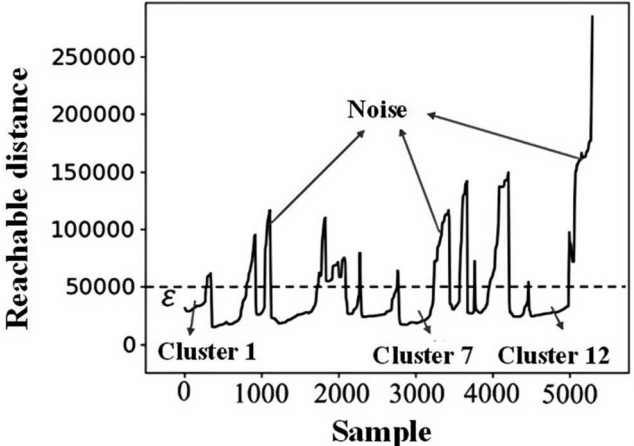

Fig.2. The result of OPTICS

Since the ship trajectory consists of multiple trajectory points, a similarity function is needed to convert the ship trajectory into sample points. In this paper, Hausdorff distance is used to measure the similarity of trajectories with the following formula:

$$dHau(T_a, T_b) = max \left\{ \begin{array}{l} \sup\limits_{a \in M} \left[ \inf\limits_{b \in N} (dis(ab)) \right], \\ \sup\limits_{b \in N} \left[ \inf\limits_{a \in M} (dis(ba)) \right] \end{array} \right\} \quad (2)$$

where $dis(ab)$ and $dis(ba)$ are one-way distances between trajectory points $a$ and trajectory points $b$; $\sup\limits_{a \in M} \left[ \inf\limits_{b \in N} (dis(ab)) \right]$ and $\sup\limits_{b \in N} \left[ \inf\limits_{a \in M} (dis(ba)) \right]$ are one-way distances between trajectory $T_a$ and trajectory $T_b$.

The ship trajectory data can be clustered by OPTICS algorithm to obtain many trajectory clusters. Each trajectory cluster represents a traffic pattern, which has its own navigational regularity.

Secondly, an approach based on course change characteristics and probabilistic statistics is used to mine the navigational regularity in the above trajectory clusters, so as to identify abnormal navigation behaviors. For capturing course changes more clearly, derives the course data to obtain the rate of turn (ROT) data. ROT data is able to represent her altering course more visually. The behavior of altering course is shown as an increasing (right turn) or decreasing (left turn) in the course data. And it is expressed as a complete convex function (right turn) or concave function (left turn) in ROT data.

By pre-processing the data, the ship trajectory is denoted as $T^{[t_0,t_n]} = \{p^{(t_0)}, ..., p^{(t_n)}\}$, the course data is denoted as $c^{[t_0,t_n]} = \{c^{(t_0)}, ..., c^{(t_n)}\}$, and the derived ROT data is $r^{[t_0,t_n]} = \{r^{(t_0)}, ..., r^{(t_n)}\}$. To identify the behaviour of altering course, the local extreme moments of the ROT are obtained by the following equation:

$$\left\{ r^{t_f} \mid \frac{dr^{(t_f)}}{dt} = 0 \ and \ \frac{d^2 r^{(t_f)}}{dt^2} \neq 0, \quad t_f \epsilon [t_0, t_n] \right\} \quad (3)$$

Each local extreme moment $t_f$ corresponds to a behaviour of altering course, and the altering amplitude corresponding to each altering behaviour can be solved by the following equation:

$$\Delta c(t_f) = c^{(t_e)} - c^{(t_s)} \quad (4)$$

where $\Delta c(t_f)$ means the altering amplitude at the local extreme moment $t_f$; $t_e$ and $t_s$ mean the moments when ROT is equal to 0 on both sides of the moment of $t_f$, as shown by the black dots in Figure 3.

However, in reality, her course also is affected by maneuvering, wind, current, wave and other environmental elements. For declining the influence of environmental elements to the behavior of altering course, some behaviors of altering is excluded by setting altering amplitude threshold ($\tau_c$). When $\Delta c(t_f) \geq \tau_c$, considering the altering course is caused by ship steer. When $\Delta c(t_f) < \tau_c$, considering the altering course is caused by environment.

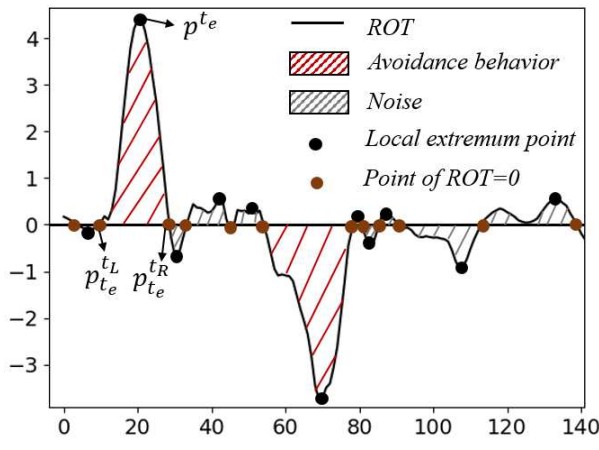

Fig.3. The ROT of OS

By $\tau_c$ and Eq. (7), the behavior of altering course can be extracted from trajectory. In this voyage, the sub-trajectories of other ship which came from the same cluster are extracted and denoted as $T_{ts}^{[t_s,t_e]} = \left\{ \left( p_{ts_i}^{(t_s)}, ..., p_{ts_i}^{(t_e)} \right) \mid i = 1,2, ..., n \right\}$. By Eq. (4), the altering magnitude ($\Delta c$) is calculated separately. The set of altering magnitudes for this part of voyage can be obtained:

$$C = \{\Delta c_i \mid i = 1,2, ..., n\} \quad (5)$$

The probability density distribution of this set of altering magnitudes was calculated using probability statistics (PDF):

$$f(x) = \frac{1}{nh} \sum_{i=1}^{n} K\left( \frac{x - \Delta c_i}{h} \right) \quad (6)$$

where $f(x)$ means the estimated probability density function; $K$ means the kernel function; $h$ means the bandwidth parameter; $\Delta c_i$ means a data point in the set of altering magnitude.

Through observation, it is found that the altering magnitude distribution conforms to Gaussian distribution, so Gaussian distribution is adopted as the kernel function:

$$K(x, x_i) = \frac{1}{\sqrt{2\pi}h} exp\left(-\frac{(x - x_i)^2}{2h^2}\right) \qquad (7)$$

By the above equations, the probability distribution curve of the altering magnitude for this part of trajectory can be obtained. As shown in Figure 4, abnormal behavior of altering course can be identified by setting $\tau_p$

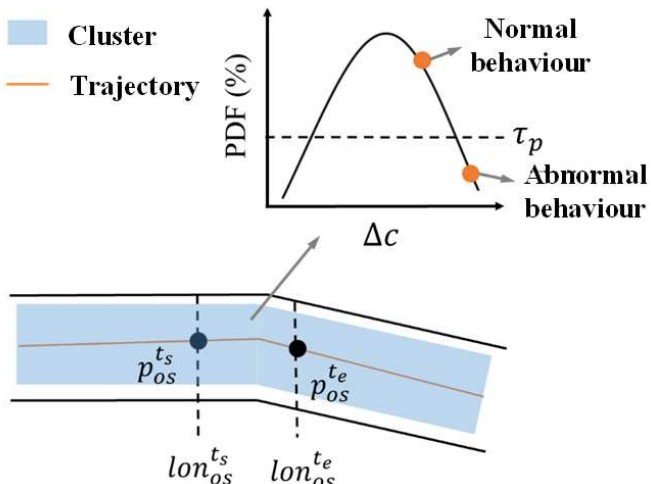

Fig.4. Abnormal behavior Identification

2)avoidance behaviors identification

In this section, avoidance behavior will be identified from the behavior of altering course by two steps.

The first step is to analyze the character of avoidance behavior to extract encounter situation with avoidance behaviors. Usually a whole process of avoiding collision consists of avoidance phase and resumption phase. So, the single abnormal behavior extracted in the last section is not a complete avoidance behavior. For forming a complete ship avoidance behavior, it is necessary to combine two abnormal behaviors. In order to judge the whole avoidance behavior, the study matches abnormal behaviors based on the following two principles:

1) The avoidance phase and the resumption phase shall be continuous or nearly continuous:

$$\left|t_s^j - t_e^i\right| \le \tau_t \qquad (8)$$

where $t_s^j$ means the start moment of the $j$th abnormal behavior; $t_e^i$ means the end moment of the $i$th abnormal behavior; $\tau_t$ is the time proximity threshold.

2) In the avoidance phase and the resumption phase, avoidance behavior should be opposite or close to opposite, and avoidance magnitude should add up to close to zero:

$$\left|\Delta c^i + \Delta c^j\right| \le \tau_{\Delta c} \qquad (9)$$

where $\Delta c^i$, $\Delta c^j$ means the altering amplitude of $i$th abnormal behavior and $j$th abnormal behavior; $\tau_{\Delta c}$ means the avoidance amplitude tolerance.

If two abnormal behaviors $T_{os}^{[t_s^i, t_e^i]}$, $T_{os}^{[t_s^j, t_e^j]}$ are satisfied with Eq. (8) and Eq. (9), this part of voyage, $\left(p_{os}^{(t_s^i)}, p_{os}^{(t_e^j)}\right)$ is considered as the whole avoidance behavior.

However, above avoidance behavior does not focus on all of near vessels. Thus, the second step of identifying avoidance behavior is to find out exactly the vessel which makes own vessel to alter course.

Relative speed is an important parameter to reflect directly the relationship between two vessels. When vessel takes some avoidance actions, relative speed will change obviously. According to it, avoidance intentions can be estimated. In the two-ship encounter scenario $\left(T_{os}^{[t_s, t_e]}, T_{ts}^{[t_s, t_e]}\right)$, $p_{os}^{(t_s)}$ and $p_{ts}^{(t_s)}$ mean the initial track points of own ship and target ship. Trend of relative motion of the two vessels is calculated by the following equation:

$$c_d = c_{rv} - c_{rb} \qquad (10)$$

where $c_{rv}$ means the angle of combined velocity of the two ships; $c_{rb}$ means the angle of the target ship relative to own ship.

This research analyses the avoidance intentions based on the following principles:

1) $c_d$ is positive or negative to represent avoidance strategy.

2)The absolute value of $c_d$ represents avoidance amplitude.

Assuming at the moment of $t_s$ own ship starts to alter course for avoiding collision and altering angle is $\Delta c$. After altering, the relative motion trend of the two vessels can be estimated by the following equation:

$$c_d' = c_{rv}' - c_{rb}' \qquad (11)$$

$$c_{rb}' = arctan\left(\frac{\Delta x}{\Delta y}\right) - (c_o + \Delta c) \qquad (12)$$

$$c_{rv}' = arctan\left(\frac{v_t sinc_t - v_{os}sin(c_o + \Delta c)}{v_t cosc_t - v_{os}cos(c_o + \Delta c)}\right) - (c_o + \Delta c)(13)$$

where $\Delta x$ and $\Delta y$ mean difference in geographic location of the two ships; $(c_o, v_o)$ and $(c_t, v_t)$ mean own ship's and target ship's speed and course.

According to compare $c_d$ with $c_d'$, estimate the intention for avoiding collision:

1) $sgn(c_d' \cdot c_d) < 0$ indicates that the avoidance strategy is reversed. At the beginning, own ship choice passing target ship from her foreword becomes passing from her stern;

2) $sgn(c_d' \cdot c_d) \ge 0$ and $|c_d'| - |c_d| \ge 0$ indicate that avoidance strategy never changes, meanwhile, the avoidance amplitude is increasing;

3) $sgn(c_d' \cdot c_d) \ge 0$ and $|c_d'| - |c_d| < 0$ indicate that avoidance strategy never changes, but the avoidance amplitude is declining.

Table I. AVOIDANCE INTENTION UNDER DIFFERENT CONDITIONGS

| $c_{rb}$ | $c_d$ | $c'_d$ | Avoidance Intention |
|---|---|---|---|
| first quadrant/ forth quadrant | $c_d \geq 0$ | $c'_d > c_d$ | Substantially pass from astern |
| | | $0 \leq c'_d \leq c_d$ | Marginally pass from astern |
| | | $c'_d < 0$ | Pass from foreword |
| | $c_d < 0$ | $c'_d < c_d$ | Substantially pass from foreword |
| | | $0 \geq c'_d \geq c_d$ | Marginally pass from foreword |
| | | $c'_d > 0$ | Pass from astern |
| second quadrant/ third quadrant | $c_d \geq 0$ | $c'_d > c_d$ | Substantially pass from foreword |
| | | $0 \leq c'_d \leq c_d$ | Marginally pass from foreword |
| | | $c'_d < 0$ | Pass from astern |
| | $c_d < 0$ | $c'_d < c_d$ | Substantially pass from astern |
| | | $0 \geq c'_d \geq c_d$ | Marginally pass from astern |
| | | $c'_d > 0$ | Pass from foreword |

According to the above table, avoidance intention is able to be judged. But avoidance intention is categorized into avoidance strategy and magnitude. The avoidance strategy and magnitude are characterized by the change of the relative bearing and the change of the DCPA with the target vessel, respectively, in order to facilitate the matching of the actual encounter process.

For example, $c_{rb}$ is located in the first quadrant at the initial moment. In the encounter situation, if the avoidance strategy is that own ship passes target ship from astern, the target's bearing will across first quadrant and second quadrant. It can be characterized by the following equation:

$$\exists t \rightarrow B(t) - B(t-1) \geq 180° \qquad (14)$$

where $B(t)$ means target ship's bearing at the moment of $t$.

If the avoidance strategy is that own ship passes target ship from foreword, the target's ship will across first quadrant and forth quadrant. The carve of changing bearing shows a general increasing trend. The overall trend of changing bearing can be characterized by a linear fit:

$$y = mx + b \qquad (15)$$

$$m = \frac{n \sum_{i=1}^{n}(x_i y_i) - \sum_{i=1}^{n} x_i \sum_{i=1}^{n} y_i}{n \sum_{i=1}^{n} x_i^2 - (\sum_{i=1}^{n} x_i)^2} \qquad (16)$$

$$b = \frac{\sum_{i=1}^{n} y_i - m \sum_{i=1}^{n} x_i}{n} \qquad (17)$$

where $m$ means the slope of the linear regression.

Except the avoidance strategy, the avoidance amplitude also be considered. DCPA is an important metric for evaluating the relative motion of two vessels. Therefore, DCPA can be used to characterize the avoidance amplitude:

$$\Delta DCPA = |DCPA(c_0 + \Delta c)| - |DCPA(c_0)| \qquad (18)$$

$$DCPA = d_r * |sin(c_{rv} - c_{rb})| \qquad (19)$$

where $DCPA(c_0 + \Delta c)$ means the distance to closest point of approach when avoidance behavior is $c_0 + \Delta c$; $DCPA(c_0)$ means the distance to closest point of approach when avoidance behavior is $c_0$; $d_r$ means the relative distance between two vessels.

Following Table II lists the avoidance characteristics under different avoidance intentions

Table II. CHARACTER OF AVOIDANCE INTENTION

| $c_{rb}$ | $c_d$ | Avoidance Intention | Characteristics of Intended Avoidance Strategy | Characteristics of Intended Avoidance amplitude |
|---|---|---|---|---|
| first quadrant/ forth quadrant | $c_d \geq 0$ | Substantially pass from astern | $\exists t \rightarrow B(t) - B(t-1) \geq 180°$ | $\Delta DCPA > 0$ |
| | | Marginally pass from astern | | $\Delta DCPA < 0$ |
| | | Pass from foreword | | / |
| | $c_d < 0$ | Substantially pass from foreword | $m > 0$ | $\Delta DCPA > 0$ |
| | | Marginally pass from foreword | | $\Delta DCPA < 0$ |
| | | Pass from astern | $\exists t \rightarrow B(t) - B(t-1) \geq -180°$ | / |
| second quadrant/ third quadrant | $c_d \geq 0$ | Substantially pass from foreword | $m < 0$ | $\Delta DCPA > 0$ |
| | | Marginally pass from foreword | | $\Delta DCPA < 0$ |
| | | Pass from astern | | / |
| | $c_d < 0$ | Substantially pass from astern | $\exists t \rightarrow B(t) - B(t-1) \leq -180°$ | $\Delta DCPA > 0$ |
| | | Marginally pass from astern | | $\Delta DCPA < 0$ |
| | | Pass from foreword | $m < 0$ | / |

## C. Constructing Ship Collision Risk Model Based on Data Driven

The ship domain is the area that the officer on watch (OOW) has made inaccessible to other vessels in order to make sure the safety of the ship. However, it is restricted by experts' acknowledgment and experience. For constructing a proper domain which is fit for research water, a data-driven ship domain is fitted by historical AIS data. Because ruder effect on both sides are different, the safety distances on both sides also be different. As following in Figure 5, an eccentric ellipsoid ship domain model is adopted as the basic model:

$$\frac{\left((x + \Delta x) \cdot cos\alpha + (y + \Delta y) \cdot sin\alpha\right)^2}{b^2} +$$

$$\frac{\left((y + \Delta y) \cdot cos\alpha - (x + \Delta x) \cdot sin\alpha\right)^2}{a^2} = 1 \qquad (20)$$

where $a$ and $b$ mean the long and short semi-axes of the ellipse; $\alpha$ means course; $\Delta x$ and $\Delta y$ mean the offset between the real ship and the virtual ship.

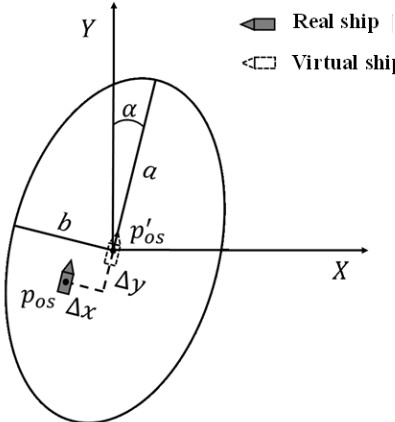

Figure 5 Eccentric elliptic ship domain

In this research, nearby ships are divided into four regions according to the relative bearing, and then the kernel density estimation can be used to obtain the probability distribution function of ship spacing in each region:

$$f(x) = \frac{1}{nh} \sum_{i=1}^{n} K\left(\frac{x - d_i}{h}\right) \qquad (21)$$

$$K(x, x_i) = \frac{1}{\sqrt{2\pi h}} exp\left(-\frac{(x - x_i)^2}{2h^2}\right) \qquad (22)$$

where $f(x)$ means the estimated probability density function; $K$ means the kernel function; $h$ means the bandwidth parameter; $d_i$ means the distance between own ship and the $i$th vessel.

By setting probability threshold $\tau_p$, the distance between two vessels is calculated:

$$\tau_p = \int_0^{d_p} f(x)dx \qquad (23)$$

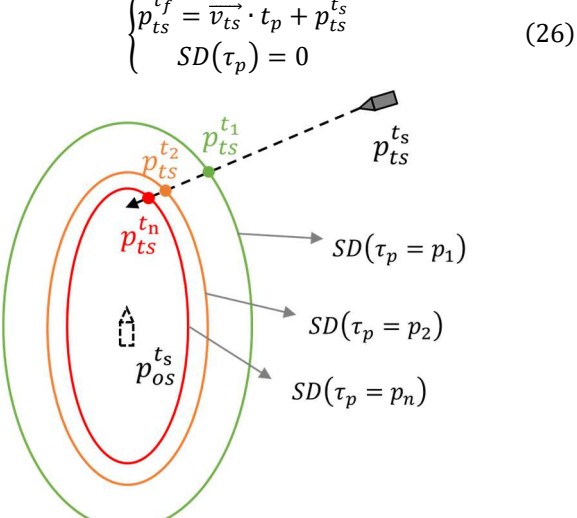

Figure 6 Diagram for solving ship spacing parameter

By Eq. (21) and Eq. (23), portside spacing value ($d_p^l$), starboard spacing value ($d_p^r$), foreword spacing value ($d_p^f$), aft spacing value ($d_p^b$) are obtained. On this basis, the long and short semi-axis parameters of the eccentric elliptical ship domain can be calculated by the following equation:

$$\begin{cases} a = \dfrac{d_p^l + d_p^r}{2} \\ b = \dfrac{d_p^f + d_p^b}{2} \end{cases} \qquad (24)$$

$$\begin{cases} \Delta x = \dfrac{d_p^r - d_p^l}{2} \\ \Delta y = \dfrac{d_p^f - d_p^b}{2} \end{cases} \qquad (25)$$

However, detecting the collision risk requires not only a proper ship domain, but also a specifying time threshold ($\tau_t$). There is a risk of collision between the two vessels means that within $\tau_t$ the target vessel enters the own vessel's domain. According to lots of historic AIS data, $\tau_p$ and $\tau_t$ can be extracted.

As shown in Figure 7, the ellipses from outside to inside are $\tau_p$ for values $p_1$ to $p_n$. $t_p$, the time that the target ship enters the ship domain of different scales can be calculated separately by jointly solving the following equation:

$$\begin{cases} p_{ts}^{t_f} = \vec{v_{ts}} \cdot t_p + p_{ts}^{t_s} \\ SD(\tau_p) = 0 \end{cases} \qquad (26)$$

Figure 7 Diagram for changing ship domain

The optimal ship domain scale should make the collision time distribution more consistent with a Gaussian distribution. The compliance of the collision time distribution with the Gaussian distribution can be estimated by the KL dispersion:

$$D_{KL}(P\|Q) = \int_{-\infty}^{\infty} P(x) \, log\frac{P(x)}{Q(x)} dx \qquad (27)$$

$$Q(x) = \frac{1}{\sigma\sqrt{2\pi}} exp\left(-\frac{(x - \mu)^2}{2\sigma^2}\right) \qquad (28)$$

where $P(x)$ means the density function of the collision time distribution; $\mu$ and $\sigma$ mean average and standard deviation of collision time set.

The parameter corresponding to $\tau_p$ with the smallest KL dispersion is taken as the ship domain parameter for the study waters. And $t_p$ under this vessel domain is determined by setting a temporal probability parameter $\omega_t \epsilon(0, 1)$:

$$t_p = \omega_t \cdot max(P(x)) \qquad (29)$$

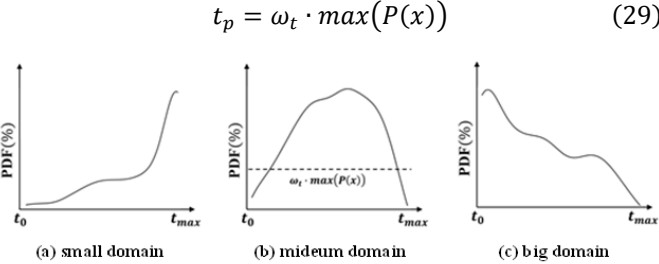

(a) small domain     (b) miedum domain     (c) big domain

Figure 8 Collision time distribution curves at different ship domains

## III. CASE STUDY

### A. Case Design

For verifying the feasibility and effectiveness of this method, a part area of the mouth of Yangtze River is chosen as the experimental water in this research. The experimental waters range in longitude from 122.3°E to 122.7°E and in latitude from 30.9°N to 31.2°N, as shown in Figure . The experimental water has three major shipping routes and two crossing areas: The South Slot Import and Export Route, the North Slot Import and Export Route, and the Outer Route. This study collected AIS data for a period of one year from March 2019 to March 2020, aiming to mine the traffic patterns in this water and identify ship avoidance behaviors, so as to construct a database of ship encounter scenarios

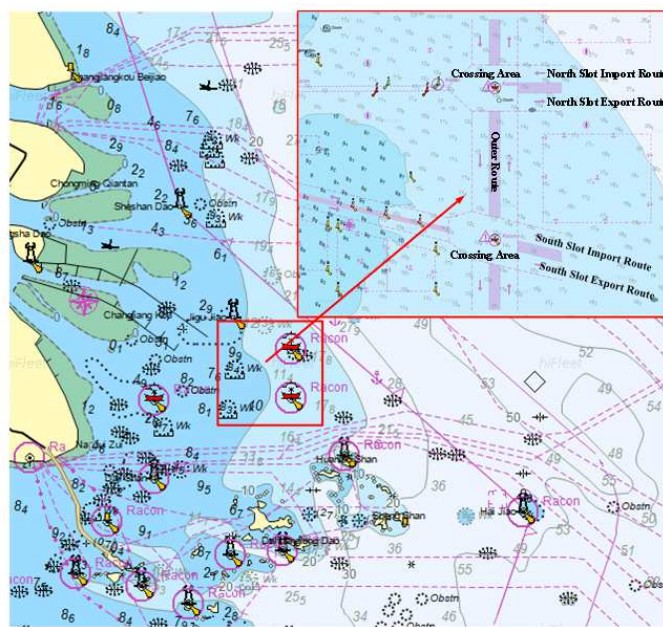

Figure 8. The experimental water

Firstly, ship AIS trajectory in experiential water is clustered by the OPTICS algorithm. Subsequently, an approach based on course change characteristics and probabilistic statistics is used to identify abnormal navigation behaviors. According to related parameters, the whole process of avoiding collision and

avoidance behaviors are filtered out as dataset. And then, based on the dataset filtered, this study separately identifies the important parameters of collision risk models based on ship domain. Finally, setting some experiment verifies the feasibility and effectiveness of the method proposed in this paper.

### B. The Result of Extracting Encounter Scenario

Figure 9 shows the initial ship trajectories after the pre-processing process of decoding, cleaning and interpolation. The initial trajectory is able to show the basic outline of the traffic patterns in the experimental water, however, it cannot clearly show the specific traffic patterns of the three major shipping routes and two intersection areas. In order to analyze and understand these complex traffic patterns more deeply, the OPTICS clustering algorithm is used to cluster the ship trajectories as shown in Figure 10.

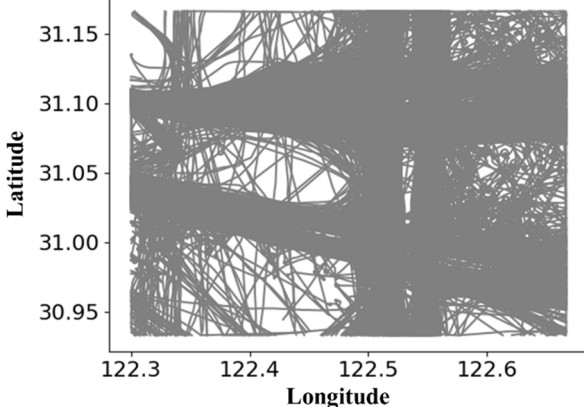

Figure 9 Initial vessel trajectory data in study water

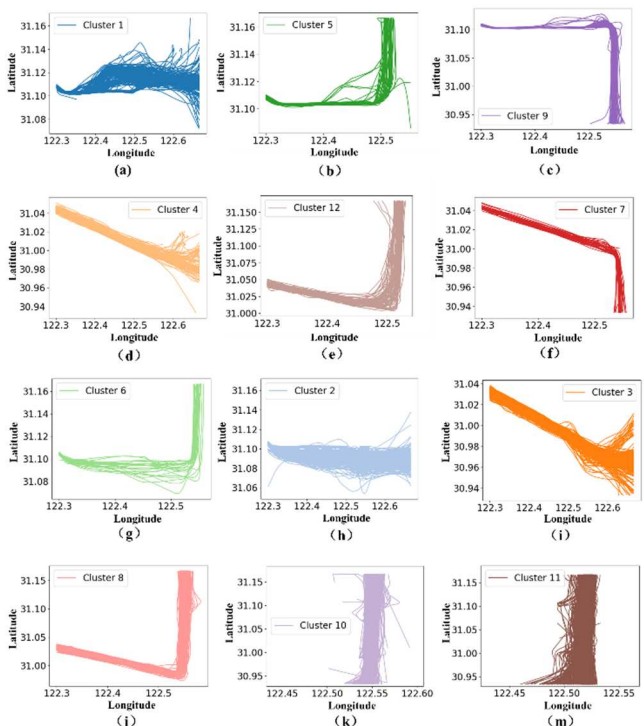

Figure 10 Cluster of 12 types of ship trajectories by OPTICS

As shown in Figure10, take the traffic patterns (e) as example. Firstly, the course of this trajectory is extracted. And then, the ROT data is derived and Eq. (3) and Eq. (4) are used to solve the local extreme points of ROT and the altering amplitude of each extreme point corresponding to the behavior of the ship. Secondly, the altering amplitude threshold $\tau_c=5°$ is set to filter out the altering behaviors. As shown in Figure 11 shows that there are five altering behaviors with the amplitude greater than 5° in this trajectory.

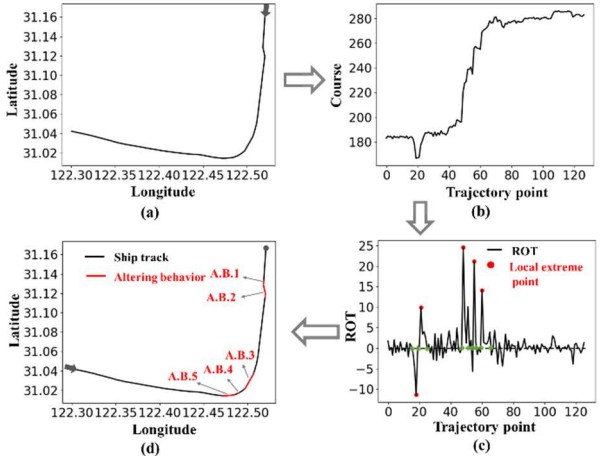

Figure 11 The process of extracting altering behavior

After extracting the altering behaviors from the ship trajectories, this study separately calculates the latitude and longitude intervals corresponding to each altering behavior, and the calculation results are shown in Table III. Then, the altering amplitude $\Delta c$ of other ships belonging to the traffic pattern (e) in this voyage is solved to obtain the altering amplitude set $C$. Secondly, the probabilistic statistic is used to obtain the probability of each altering behavior in the $C$. Finally, the three altering behaviors below the pre-set probability threshold $\tau_p=0.015$ are identified as abnormal behaviors.

Table III. ABNORMAL BEHAVIOR IDENTIFICATION STATISTICS

| Altering Behavior | Altering amplitude | Probability | Consequence |
|---|---|---|---|
| 1 | -14° | 0.008% | Abnormal |
| 2 | 12° | 0.012% | Abnormal |
| 3 | 21° | 0.0135% | Abnormal |
| 4 | 12° | 0.022% | Normal |
| 5 | 21° | 0.0168% | normal |

Setting $\tau_t$ is 102 and $\tau_{\Delta c}$ is 5°. Based on the principle of 2.1.2, a complete avoidance behavior is successfully matched from the above three abnormal behaviors.

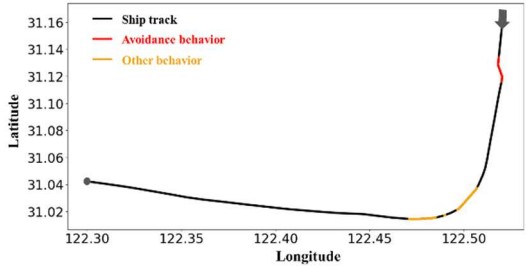

Figure 12 Avoidance behavior extracted

According to the Table I and Table II, ship avoidance intention is analyzed. Its intention is own vessel choose to pass target vessel form her astern

## C. Constructing Ship Collision Risk Model Based on Data Driven

By analyzing the avoidance behaviors and other AIS data, a more accurate and practical ship domain model can be effectively fitted. It is highly feasible and effective in the study waters. According to Eq. (23), ship domain parameters corresponding to different $\tau_p$ is obtained. As $\tau_p$ increases, the long and short semi-axis radius of the ship domain increase. However, too large a domain leads to a decrease in the accuracy of the collision risk model. Figure 13 shows the corresponding ship domain parameters for $\tau_p$ ranging from 0.005 to 0.2.

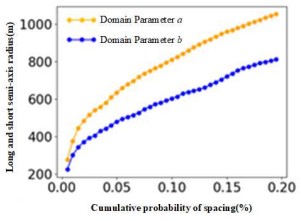 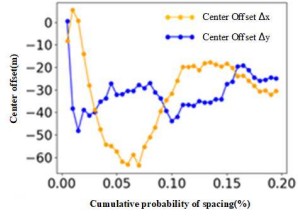

(a) The parameter of long and short semi-axis radius    (b) The parameter of center offset

Figure 13 The result of ship domain parameter identification

Another important parameter of the collision risk model based on the ship domain is the collision time for the target ship to enter the ship domain $\tau_t$. In order to extract $\tau_t$ suitable for the study waters, this experiment analyze the collision time under different scale domains. Table IV shows the KL dispersion of $\tau_p$ and $\tau_t$.

Table IV KL DISPERSION of $\tau_p$

| Parameter | $\tau_p = 0.015$ | $\tau_p = 0.02$ | $\tau_p = 0.04$ | $\tau_p = 0.2$ |
|---|---|---|---|---|
| $a$ | 444 | 484 | 578 | 1062 |
| $b$ | 341 | 371 | 440 | 820 |
| $\Delta x$ | -48 | -38 | -33 | -25 |
| $\Delta y$ | -8 | -14 | -54 | -27 |
| KL | 0.0305 | 0.0243 | 0.0221 | 0.0327 |

As shown in Figure 14, the curve has the lowest KL dispersion at $\tau_p = 0.04$, which means that the corresponding ship domain parameters are the most suitable for the experiential water. Then, by setting the collision time probability threshold $\tau_t$, the ship collision risk model based on the ship domain model can be obtained.

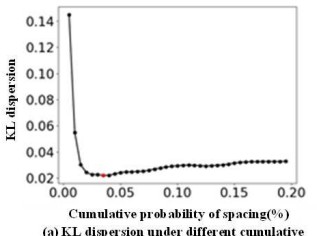 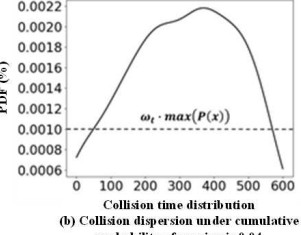

(a) KL dispersion under different cumulative probability of spacing    (b) Collision dispersion under cumulative probability of spacing is 0.04

Figure 14 Results of parameter identification in the study water

As shown in Figure 15, risk model accuracy for different $\omega_t$ tested by test set. The result indicates that the accuracy

decreases with the increase of $\omega_t$, so $\omega_t$ should not be set too high when using this model for risk detection.

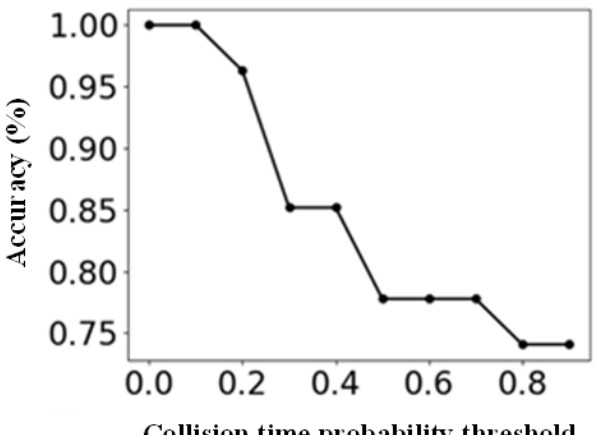

Figure 15 Accuracy of collision risk model based on the ship domain

## IV. Conclusion

The research on extracting the important parameters of collision risk model is very meaningful to ensure that ship can navigate safely. By deeply mining the information harbored in AIS data, an innovative ship collision risk models based on data-driven has been proposed. This research uses the OPTICS algorithm to cluster ship trajectories. An approach based on course change characteristics and probabilistic statistics has been used to identify abnormal navigation behaviors. Analyzing the character of avoidance behavior extract encounter situation with avoidance behaviors. Finally, the key parameters of risk collision model have been extracted by above behaviors, and the collision risk model has been constructed. In the process of modeling, this approach has considered both the AIS data and the collision avoidance behavior. This collision risk model is more accurate than conventional collision risk model.

For varying the feasibility and effectiveness of this method which is proposed in this research, one year's worth of AIS data have been used in the case study. This experiment successfully has extracted avoidance behaviors, encounter situation, and obtained the key parameter of collision risk model fitted for study water.

The approach which is proposed in the research still has some deficiencies. In the process of extracting encounter situation, there is phenomenon that a little behavior is identified as avoidance behavior by mistake. In the process of extracting parameter, the result of them are influenced extremely by the quality of AIS data. Therefore, the above flaws will be considered in future research.

## Acknowledgment

This work is supported by the Postdoctoral Fellowship Program of CPSF under Grant Number GZC20241296.

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
