# OpenReview forum: "A Data-Driven Method for Ship Collision Risk Detection in Heavy Traffic Waters"
_IEEE.org/ICIST/2024/Conference — IEEE ICIST 2024 Conference Submission_

### Official Review · Reviewer_Sm18 · 2024-08-21
**This paper proposes a novel collision risk model based on data-driven to solve the problem of poor neralization of complex collision risk model. Firstly, the main traffic  pattern is clustered by ordering points to identify the clustering  structure algorithm. Afterwards, using probability statistics and  mining the altering behaviors identify the abnormal behavior. And  then, encounter situation and avoidance behaviors are identified  by analyzing the relative motion characteristics of the two ships.  Finally, by Kullback-Leibler dispersion and kernel density  estimation methods, the key parameters are extracted from  encounter data with avoidance behaviors. The collision risk model  based on data-driven is constructed. In my opinion, there are some comments that should be considered during the revision of the manuscript which are listed below:**

**Rating:** 7
**Confidence:** 3

**Review:**

1. The layout of the article is not neat enough, e.g. ‘where 𝑚 means the slope of the linear regression’ does not need indentation characters, etc. Please check the whole article and correct them.
2. The references should be updated. Some closely related and new references should be added to show to further explain the novelty and innovation of the work.
3. This study may have different designs in comparison to some previous studies, but how are these differences significant? The author needs to highlight and clarify them in innovation point.
4. In the simulation section, the clarity of the figure is too low, it is recommended to convert the image to '.eps ' format.

---

### Official Review · Reviewer_wQXD · 2024-08-21
**accept**

**Rating:** 7
**Confidence:** 3

**Review:**

Comment: This paper proposes a novel collision risk model based on data-driven. It aims at solve the problem of poor generalization of complex collision risk model. The theory is correct and can be accepted after responding the following comments.
(1) More comprehensive literature review is needed to clarify the research gap and research motivation.
(2) On the third page, Fig.2 is repeated with Sample.
(3) The conclusion of the article suggests using the present perfect tense for description.

---

### Official Review · Reviewer_XvQy · 2024-08-22
**The paper presents a novel method to improve the accuracy of ship collision risk detection, particularly in areas with heavy maritime traffic. The research addresses the limitations of existing models, which often struggle with generalization and are influenced by subjective parameters.**

**Rating:** 8
**Confidence:** 4

**Review:**

The paper titled "A Data-Driven Method for Ship Collision Risk Detection in Heavy Traffic Waters" presents a novel method to improve the accuracy of ship collision risk detection, particularly in areas with heavy maritime traffic. The research addresses the limitations of existing models, which often struggle with generalization and are influenced by subjective parameters. The topic of this paper is interesting. Below is a list of comments that should be taken into account further when revising the paper.

1.	There are a few typos in this paper which should be corrected. And there are some notions missed. Please make some corrections.
2.	The background and motivation of research on Data-Driven Method can be updated and more explained in Section I.
3.	More comments on figures should be given in the revision. Please add the necessary comments for Figures.

---

### Decision · Program_Chairs · 2024-09-06

Accept (Oral)